# Group invariants for Feynman diagrams

Idrish Huet[1], Michel Rausch de Traubenberg[2] and Christian Schubert[3⋆]

**1** Facultad de Ciencias en Física y Matemáticas, Universidad Autónoma de Chiapas
Ciudad Universitaria, Tuxtla Gutiérrez 29050, Mexico
**2** Université de Strasbourg, CNRS, IPHC UMR7178, F-67037 Strasbourg Cedex, France
**3** Centro Internacional de Ciencias A.C. Campus UNAM-UAEM Avenida Universidad
1001 Cuernavaca Morelos Mexico C.P. 62100 * christianschubert137@gmail.com

December 18, 2022

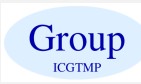

## Abstract

**It is well-known that the symmetry group of a Feynman diagram can give important information on possible strategies for its evaluation, and the mathematical objects that will be involved. Motivated by ongoing work on multi-loop multi-photon amplitudes in quantum electrodynamics, here I will discuss the usefulness of introducing a polynomial basis of invariants of the symmetry group of a diagram in Feynman-Schwinger parameter space.**

## 1 Introduction: Schwinger parameter representation of Feynman diagrams

The most universal approach to the calculation of Feynman diagrams uses Feynman-Schwinger parameters $x_i$, introduced through the exponentiation of the (Euclidean) scalar propagator,

$$\frac{1}{p^2 + m^2} = \int_0^\infty dx \, \mathrm{e}^{-x(p^2+m^2)}$$

For scalar diagrams, one finds the following universal structure for an arbitrary graph $G$ with $n$ internal lines and $l$ loops in $D$ dimensions:

$$I_G = \Gamma(n - lD/2) \int_{x_i \geq 0} d^n x \, \delta\Big(1 - \sum_{i=1}^n x_i\Big) \frac{\mathcal{U}^{n-(l+1)\frac{D}{2}}}{\mathcal{F}^{n-lD/2}}$$

$\mathcal{U}$ and $\mathcal{F}$ are polynomials in the $x_i$ called the first and second Symanzyk (graph) polynomials. There exist graphical methods for their construction.

For more general theories (involving not only scalar particles) the same graph will involve, in addition to these two polynomials, also a *numerator polynomial* $\mathcal{N}(x_1, \ldots, x_n)$.

## 2 Symmetries of Feynman diagrams

Many Feynman diagrams possess a non-trivial symmetry group, generated by interchanges of the internal lines that leave the topology of the graph unchanged. Then all its graph

polynomials must be invariant under the natural action of the group on the set of polynomials $\mathbb{R}[x_1, \ldots, x_n]$, $g.P(X) \equiv P(g.X)$.

A nice example is the $l - loop\ banana\ graph$ shown in Fig. 1.

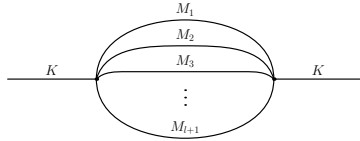

Figure 1: $l - loop$ banana graph.

This graph has full permutation symmetry in all the internal lines, so the symmetry group is $S_{l+1}$, and its graph polynomials must be symmetric functions of $x_1, \ldots, x_{l+1}$. As is well-known, this implies that they can be rewritten as polynomials in the *elementary symmetric polynomials* $S_1, \ldots, S_n$.

Perhaps less known is that this generalizes to the case of a general symmetry group as follows [1]:

**Theorem 1**: Let $G$ be a finite group and let $\Gamma$ be an $n-$dimensional (real) representation.

1. There exist $n = \dim(\Gamma)$ algebraically invariant polynomials $P_1, \cdots, P_n$, called the *primitive invariants*, such that the Jacobian $\frac{\partial(P_1,\ldots,P_n)}{\partial(x_1,\ldots,x_n)} \neq 0$.

2. Denote $d_k = \deg(P_k)$ and $\mathcal{R} = \mathbb{R}[P_1, \cdots, P_n]$ the subalgebra of polynomial invariants generated by the primitive invariants.

3. There exists $m = d_1 \cdots d_n/|G|$ secondary invariants polynomials $S_1, \cdots, S_m$.

4. The subalgebra of invariants $\mathbb{R}[x_1, \cdots, x_n]^\Gamma$ is a free $\mathcal{R}-$module with basis $(S_1, \cdots, S_m)$. In particular this means that any invariant $I \in \mathbb{R}[x_1, \cdots, x_n]^\Gamma$ can be uniquely written as $I = \sum_{i=1}^m f_i(P_1, \cdots, P_n)S_i$, where $f_i(P_1, \cdots, P_n), i = 1, \cdots, m$ belong to $\mathcal{R}$, *i.e.*, are polynomials in $(P_1, \cdots, P_n)$.

There exist computer algebra systems for the computation of $P_1, \ldots, P_n$ such as SINGULAR [2].

# 3   The Euler-Heisenberg Lagrangian at one loop

In 1936 Heisenberg and Euler obtained their following famous representation of the one-loop QED effective Lagrangian in a constant field (" Euler-Heisenberg Lagrangian")

$$\mathcal{L}^{(1)}(a,b) \;=\; -\frac{1}{8\pi^2}\int_0^\infty \frac{dT}{T^3}\,\mathrm{e}^{-m^2T}\left[\frac{(eaT)(ebT)}{\tanh(eaT)\tan(ebT)} - \frac{e^2}{3}(a^2 - b^2)T^2 - 1\right] \quad (1)$$

Here $m$ is the electron mass, and $a, b$ are the two invariants of the Maxwell field, related to $\mathbf{E}, \mathbf{B}$ by $a^2 - b^2 = B^2 - E^2, \quad ab = \mathbf{E} \cdot \mathbf{B}$.

This effective Lagrangian hold the information on the QED $N$ - photon amplitudes in the low-energy limit where all photon energies are small compared to the electron mass, $\omega_i \ll m$. It corresponds to the Feynman diagrams shown in Fig. 2.

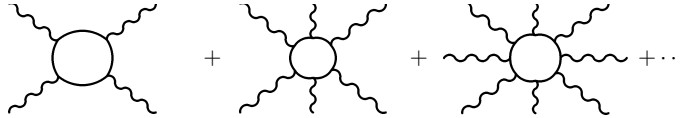

Figure 2: Feynman diagrams corresponding to the Euler-Heisenberg Lagrangian. The photon legs are in the low-energy limit.

For the extraction of the amplitudes from the effective Lagrangian, one expands it in powers of the Maxwell invariants,

$$\mathcal{L}(a,b) = \sum_{k,l} c_{kl}\, a^{2k} b^{2l} \tag{2}$$

then fixes a helicity assignment and uses spinors helicity techniques [3].

## 4  Imaginary part of the effective action

Except for the purely magnetic case where $b = 0$, the proper-time integral in (1) has poles on the integration contour at $ebT = k\pi$ which create an imaginary part. For the purely electric case one gets [4]

$$\mathrm{Im}\mathcal{L}^{(1)}(E) \;=\; \frac{m^4}{8\pi^3}\beta^2 \sum_{k=1}^{\infty} \frac{1}{k^2}\, \exp\left[-\frac{\pi k}{\beta}\right]$$

($\beta = eE/m^2$). We note:

- The $k$th term relates to coherent creation of $k$ pairs in one Compton volume.

- In the weak-field limit $\beta \ll 1$ the terms with $k \geq 2$ can be neglected.

- $\mathrm{Im}\mathcal{L}(E)$ depends on $E$ non-perturbatively (non-analytically), which is consistent with Sauter's [5] interpretation of pair creation as vacuum tunnelling (Fig. 3).

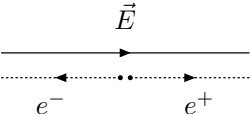

Figure 3: Pair creation by an external field as vacuum tunnelling.

## 5  Beyond one loop

The two-loop correction to the Euler-Heisenberg Lagrangian due to one internal photon exchange (Fig. 4) has been analyzed [6–8], and turned out to contain important information on the Sauter tunnelling picture [9], on-shell versus off-shell renormalization [6, 10], and the asymptotic properties of the QED photon S-matrix [11].

It leads to rather intractable two-parameter integrals. However, in the electric case its imaginary part $\mathrm{Im}\mathcal{L}^{(2)}(E)$ permits a decomposition analogous to Schwinger's (3) [9]. For single-pair production, this is now interpreted as a a tunnelling process where, in the

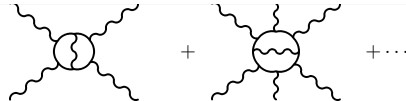

Figure 4: Feynman diagrams contributing to the 2-loop EHL.

process of turning real, the electron-positron pair is already interacting at the one-photon exchange level.

Even for the imaginary part no completely explicit formulas are available. However, it simplifies dramatically in the weak-field limit, where it just becomes an $\alpha\pi$ correction to the one-loop contribution:

$$\mathrm{Im}\mathcal{L}^{(1)}(E) + \mathrm{Im}\mathcal{L}^{(2)}(E) \overset{\beta\to 0}{\sim} \frac{m^4\beta^2}{8\pi^3}\left(1 + \alpha\pi\right)\mathrm{e}^{-\frac{\pi}{\beta}}$$

This suggests that higher loop orders might lead to an exponentiation, and indeed Lebedev and Ritus [9] provided strong support for this hypothesis by showing that, assuming that

$$\mathrm{Im}\mathcal{L}^{(1)}(E) + \mathrm{Im}\mathcal{L}^{(2)}(E) + \mathrm{Im}\mathcal{L}^{(3)}(E) + \ldots \overset{\beta\to 0}{\sim} \frac{m^4\beta^2}{8\pi^3}\exp\left[-\frac{\pi}{\beta} + \alpha\pi\right] = \mathrm{Im}\mathcal{L}^{(1)}(E)\,\mathrm{e}^{\alpha\pi}$$

then the result can be interpreted in the tunnelling picture as the corrections to the Schwinger pair creation rate due to the pair being created with a negative Coulomb interaction energy

$$m(E) \approx m + \delta m(E), \quad \delta m(E) = -\frac{\alpha}{2}\frac{eE}{m}$$

Moreover, the resulting field-dependent mass-shift $\delta m(E)$ is identical with the *Ritus mass shift*, originally derived by Ritus in [13] from the crossed process of one-loop electron propagation in the field (Fig. 5).

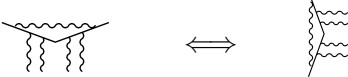

Figure 5: Photon-corrected pair-creation vs. electron propagation in the field.

Unbeknownst to the authors of [9], for scalar QED the corresponding conjecture had already been established two years earlier by Affleck, Alvarez and Manton [12] using Feynman's worldline path integral formalism and a semi-classical *worldline instanton* approximation.

Diagrammatically, we note the following features of the exponentiation formula (see Fig. 6):

- It Involves diagrams with any numbers of loops and legs.

- Although not shown, also all the counter-diagrams from mass renormalization must contribute.

- It does *not* include diagrams with more than one fermion loop (those get suppressed in the weak-field limit [12]).

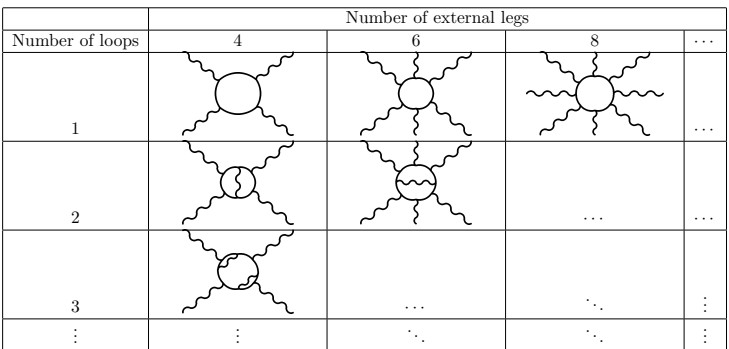

Figure 6: Feynman diagrams contributing to the exponentiation hypothesis.

- Horizontal summation produces the Schwinger exponential $\mathrm{e}^{-\frac{\pi}{\beta}}$.

- Vertical summation produces the Ritus-Lebedev/Affleck-Alvarez-Manton exponential $\mathrm{e}^{\alpha\pi}$.

# 6   QED in 1+1 dimensions

The exponentiation conjecture has so far been verified only at two loops. A three-loop check is in order, but calculating the three-loop EHL in $D = 4$ is presently hardly feasible. Motivated by work by Krasnansky [14] on Euler-Heisenberg in various dimensions, in 2010 two of the authors with D.G.C. McKeon started investigating the analogous problem in 2D QED. In [15] we used the worldline instanton method to generalize the exponentiation conjecture to the 2D case, resulting in

$$\mathrm{Im}\mathcal{L}_{2D}^{(all-loop)} \sim \mathrm{e}^{-\frac{m^2\pi}{eE}+\tilde{\alpha}\pi^2\kappa^2} \tag{3}$$

where $\kappa = m^2/(2ef)$, $f^2 = \frac{1}{4}F_{\mu\nu}F^{\mu\nu}$, and $\tilde{\alpha} = \frac{2e^2}{\pi m^2}$ is the two-dimensional analogue of the fine-structure constant. Defining the weak-field expansion coefficients in $2D$ by

$$\mathcal{L}^{(l)(2D)}(\kappa) = \frac{m^2}{2\pi}\sum_{n=1}^{\infty}(-1)^{l-1}c_{2D}^{(l)}(n)(i\kappa)^{-2n} \tag{4}$$

we then used Borel analysis to derive from (3) a formula for the limits of ratios of $l$ - loop to one - loop coefficients:

$$\lim_{n\to\infty}\frac{c_{2D}^{(l)}(n)}{c_{2D}^{(1)}(n+l-1)} = \frac{(\tilde{\alpha}\pi^2)^{l-1}}{(l-1)!} \tag{5}$$

Moreover, we calculated the 2D Euler-Heisenberg Lagrangian at one and two loops,

$$\mathcal{L}^{(1)}(f) = -\frac{m^2}{4\pi}\frac{1}{\kappa}\left[\ln\Gamma(\kappa) - \kappa(\ln\kappa - 1) + \frac{1}{2}\ln\left(\frac{\kappa}{2\pi}\right)\right] \tag{6}$$

$$\mathcal{L}^{(2)}(f) = \frac{m^2}{4\pi}\frac{\tilde{\alpha}}{4}\left[\tilde{\psi}(\kappa) + \kappa\tilde{\psi}'(\kappa) + \ln(\lambda_0 m^2) + \gamma + 2\right] \tag{7}$$

where $\tilde{\psi}(x) \equiv \psi(x) - \ln x + \frac{1}{2x}$, $\psi(x) = \Gamma'(x)/\Gamma(x)$, and the constant $\lambda_0$ comes from an IR cutoff. One finds from (6) and (7) that

$$c_{2D}^{(1)}(n) = (-1)^{n+1}\frac{B_{2n}}{4n(2n-1)} \tag{8}$$

$$c_{2D}^{(2)}(n) = (-1)^{n+1}\frac{\tilde{\alpha}}{8}\frac{2n-1}{2n}B_{2n} \tag{9}$$

Using properties of the Bernoulli numbers $B_n$ it is then easy to verify that

$$\lim_{n \to \infty} \frac{c_{2D}^{(2)}(n)}{c_{2D}^{(1)}(n+1)} = \tilde{\alpha}\pi^2$$

in accordance with (5).

# 7 Three-loop EHL in 2D: diagrams

At three loops, we face the task of computing the two diagrams shown in Fig. 7 (there are also diagrams involving more than one fermion-loop, including several that involve Gies-Karbstein tadpoles [16], but those can be shown to be subdominant in the asymptotic limit).

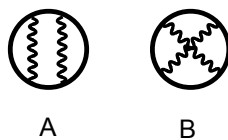

Figure 7: Three-loop diagrams contributing to the exponentiation conjecture.

Due to the super-renormalizability of $2D$ QED these diagrams are already UV finite. They suffer from spurious IR - divergences, but those can be removed by going to the *traceless gauge* $\xi = -2$ [17]. The calculation of diagram A is relatively straightforward, thus we focus on the much more substantial task of computing diagram B and its weak-field expansion coefficients.

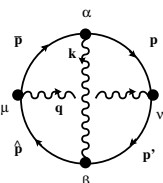

Figure 8: Parametrization of diagram B.

Introducing Schwinger parameters for this diagram as shown in Fig. 8 leads to the integral representation [17]

$$\mathcal{L}^{3B}(f) = \frac{\tilde{\alpha}^2 m^2}{128\pi} \int_0^\infty dw\, dw'\, d\hat{w}\, d\bar{w}\ I_B\ \mathrm{e}^{-a}$$

$$I_B = \frac{\rho^3}{\cosh^2 \rho w \cosh^2 \rho w' \cosh^2 \rho \hat{w} \cosh^2 \rho \bar{w}} \frac{B}{A^3 C}$$
$$-\rho \frac{\cosh(\rho \tilde{w})}{\cosh \rho w \cosh \rho w' \cosh \rho \hat{w} \cosh \rho \bar{w}} \left[ \frac{1}{A} - \frac{C}{G^2} \ln\left(1 + \frac{G^2}{AC}\right) \right]$$

where

$$B = (\tanh^2 z + \tanh^2 \hat{z})(\tanh z' + \tanh \bar{z}) + (\tanh^2 z' + \tanh^2 \bar{z})(\tanh z + \tanh \hat{z})$$
$$C = \tanh z \tanh z' \tanh \hat{z} + \tanh z \tanh z' \tanh \bar{z} + \tanh z \tanh \hat{z} \tanh \bar{z} + \tanh z' \tanh \hat{z} \tanh \bar{z}$$
$$G = \tanh z \tanh \hat{z} - \tanh z' \tanh \bar{z}$$

$(z = \rho w$ etc.$)$. Although for a three-loop diagram this is a fairly compact representation, an exact calculation is out of the question, and a straightforward expansion in powers of the external field to get the weak-field expansion coefficients turns out to create huge numerator polynomials. To deal with those, we will now take advantage of the high symmetry of the diagram.

## 8   Integration-by-parts algorithm

Introduce the operator $\tilde{d} \equiv \frac{\partial}{\partial w} - \frac{\partial}{\partial w'} + \frac{\partial}{\partial \hat{w}} - \frac{\partial}{\partial \bar{w}}$ which acts simply on the trigonometric building blocks of the integrand. Integrating by parts with this operator, it is possible to write the integrand of the $n$-th coefficient $\beta_n$ as a total derivative $\beta_n = \tilde{d}\theta_n$. Then, using once more the symmetry of the graph,

$$
\begin{aligned}
\int_0^\infty dw\,dw'\,d\hat{w}\,d\bar{w}\; \mathrm{e}^{-a}\beta_n &= \int_0^\infty dw\,d\bar{w}\,d\hat{w}\,dw'\,\tilde{d}\; \mathrm{e}^{-(w+w'+\hat{w}+\bar{w})}\theta_n \\
&= 4\int_0^\infty dw\,dw'\,d\hat{w}\; \mathrm{e}^{-(w+w'+\hat{w})}\,\theta_n|_{\bar{w}=0}
\end{aligned}
$$

The remaining threefold integrals are already of a fairly standard type.

## 9   Using the polynomial invariants of $D_4$

Diagram $B$ has the symmetries

$$
\begin{aligned}
w &\leftrightarrow \hat{w} \\
w' &\leftrightarrow \bar{w} \\
(w, \hat{w}) &\leftrightarrow (w', \bar{w})
\end{aligned}
$$

Those generate the dihedral group $D_4$. After a slight generalization to the inclusion of semi-invariants (invariants up to a sign) [18], Theorem 1 can be used to deduce that the numerator polynomials can be rewritten as polynomials in the variable $\tilde{w} = w - w' + \hat{w} - \bar{w}$ with coefficients that are polynomials in the four $D_4$ - invariants $a, v, j, h$,

$$
\begin{aligned}
a &= w + w' + \hat{w} + \bar{w} \\
v &= 2(w\hat{w} + w'\bar{w}) + (w + \hat{w})(w' + \bar{w}) \\
j &= a\tilde{w} - 4(w\hat{w} - w'\bar{w}) \\
h &= a(ww'\hat{w} + ww'\bar{w} + w\hat{w}\bar{w} + w'\hat{w}\bar{w}) + (w\hat{w} - w'\bar{w})^2
\end{aligned}
$$

These invariants are moreover chosen such that they are annihilated by $\tilde{d}$. Thus they are well-adapted to the integration-by-parts algorithm. This rewriting leads to a very significant reduction in the size of the expressions generated by the expansion in the field.

## 10   Results

In this way we obtained the first two coefficients of the weak-field expansion analytically,

$$
\begin{aligned}
\Gamma_0^B &= -\frac{3}{2} + \frac{7}{4}\zeta(3) \\
\Gamma_1^B &= -\frac{251}{120} + \frac{35}{16}\zeta(3)
\end{aligned}
$$

and five more coefficients numerically. For a definite conclusion concerning the exponentiation conjecture this is still insufficient, and the computation of further coefficients is in progress.

## 11   Outlook

- Writing Feynman graph polynomials in terms of invariant polynomials is a universal option that, to the best of our knowledge, has not previously been used, but we expect that it will be found very useful for multiloop calculations involving a large number of external legs.

- In particular, this is the case for the weak-field expansion of the QED effective Lagrangian starting from three loops (in any dimension).

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
