# Peer review of "Group invariants for Feynman diagrams"

_SciPost Physics Proceedings_

## Round 1 · Referee Report · Anonymous (Referee 1) · 2022-12-21

Strengths
1. Compact overview.
2. Well written.
Weaknesses
Slightly too much weight on the example of the Euler-Heisenberg example, while the main topic is the symmetries of Feynman diagrams.
Report
I recommend to publish this article in the proceedings with the minor modifications discussed below.
Requested changes
1. It would be useful to give a brief overview of the paper in section 1, in particular on how the QED example is embedded into the discussion of properties of Feynman integrals and that the author comes back to the latter at the end.
2. In the first sentence of section 2 the symmetries under interchanges of internal lines are emphasized. It would be useful to comment on the exchange symmetries in external lines and why they are not discussed here.
3. At the beginning of section 3 the abbreviation EHL should be introduced, which is used later on page 5.
4. At the beginning of section 8 the beta_n should be defined, i.e. of what are they the coefficients? Similarly for Gamma_n at beginning of section 10.
5. The outlook states that the method of the paper would be useful at large number of external legs. Since it is a symmetry in the internal lines, wouldn't it rather be useful for a large number of internal legs?
6. I found a few typos and similar points:
* Point 3. of Theorem 1: exit[s] and invariant[s]
* Bottom of page 2: This effective Lagrangian hold[s]
* Bottom of page 3: as [a] a tunnelling
* Section 6: Make 2D and 2D consistent (text- vs math-mode)
* Bottom of page 6: the z's in the expressions for B,C,G should be in math-mode as on the top of page 7.

---

## Editorial Decision

unknown